# Partnering with First Nations in Northern British Columbia Canada to Reduce Inequity in Access to Genomic Research

**DOI:** 10.3390/ijerph20105783

**Published:** 2023-05-10

**Authors:** Nadine R. Caron, Wilf Adam, Kate Anderson, Brooke T. Boswell, Meck Chongo, Viktor Deineko, Alexanne Dick, Shannon E. Hall, Jessica T. Hatcher, Patricia Howard, Megan Hunt, Kevin Linn, Ashling O’Neill

**Affiliations:** 1UBC Northern Medical Program and Department of Surgery, The University of British Columbia, Vancouver, BC V6T 1Z4, Canada; 2UBC Centre for Excellence in Indigenous Health, The University of British Columbia, Vancouver, BC V6T 1Z4, Canada; 3First Nations Health Authority Chair in Cancer and Wellness at UBC, UBC Health and Faculty of Medicine, The University of British Columbia, Vancouver, BC V6T 1Z4, Canada; 4Elder Advisor, Burns Lake, BC V0J 1E0, Canada; 5School of Public Health, University of Queensland, Brisbane, QLD 4067, Australia; 6Community Health Sciences, University of Northern British Columbia (UNBC), Prince George, BC V2N 4Z9, Canada; 7University of Saskatchewan, Saskatoon, SK S7N 5A2, Canada; 8Northern Biobank at the University Hospital of Northern BC, Northern Health, George, BC V2M 1S2, Canada; 9First Nations Biobank, Department of Surgery, Faculty of Medicine, Vancouver Campus, University of British Columbia, Vancouver, BC V6T 1Z4, Canada; 10UBC Faculty of Medicine, The University of British Columbia, Vancouver, BC V6T 1Z4, Canada; 11First Nations Health Authority, Northern Region, Prince George, BC V2L 5R8, Canada; 12UBC Northern Medical Program, Faculty of Medicine, University of British Columbia, Vancouver, BC V6T 1Z4, Canada; 13School of Health Sciences, University of Northern British Columbia, Prince George, BC V2N 4Z9, Canada

**Keywords:** Indigenous, First Nations, biobank, genomics, indigenous governance, cultural safety, qualitative, community engagement, trust, partnership, data sovereignty, capacity building, consent

## Abstract

Indigenous-led, culturally safe health research and infrastructure are essential to address existing inequities and disparities for Indigenous Peoples globally. Biobanking, genomic research, and self-governance could reduce the existing divide and increase Indigenous participation in health research. While genomic research advances medicine, barriers persist for Indigenous patients to benefit. In northern BC, Canada, the Northern Biobank Initiative (NBI), with guidance from a Northern First Nations Biobank Advisory Committee (NFNBAC), has engaged in consultations with First Nations on biobanking and genomic research. Key informant interviews and focus groups conducted with First Nations leaders, Elders, Knowledge Keepers, and community members established culturally safe ways of biobanking and exploring genomic research. Strong support for a Northern British Columbia First Nations Biobank (NBCFNB) that will promote choice, inclusion, and access to health research opportunities emerged. The acceptance and enthusiasm for the development of this NBCFNB and its governance table highlight the shift towards Indigenous ownership and support of health research and its benefits. With engagement and partnership, community awareness, multigenerational involvement, and support from diverse and experienced healthcare leaders, the NBCFNB will establish this culturally safe, locally driven, and critically important research priority that may serve as an example for diverse Indigenous groups when designing their unique biobanking or genomic research opportunities.

## 1. Background

Considerable progress in the field of biobanking and related genomic research has precipitated better clinical diagnoses and treatments for patients globally. Biobanks are collections of human biospecimens linked to associated health data (in addition to demographic and clinical elements, health data here may also include socio-economic, genealogical, lifestyle, and environmental data) that are collected and used under strict conditions of confidentiality, biobank participant rights, and responsible use [1]. These expanding resources, however, are seldom subject to cultural perspectives and governance guidelines. With the appropriate infrastructure, governance, and protocols in place, biobank collections can be created for the purposes of research that can be inclusive, respectful, and relevant to participants [2].

Similar to Indigenous Peoples globally, First Nations Peoples (First Nations Peoples in Canada are one of three groups of Indigenous Peoples, along with the Métis and Inuit, who have lived on and cared for the land and ancestral lineage since time immemorial). In northern British Columbia (BC), Canada faces significant challenges regarding inequitable representation, participation, and access to biobanking, associated research, and subsequent clinical applications [3,4]. Complex diseases, such as cancer, are seldom ascribed solely to malfunctions of individual genes or the influence of isolated biological factors. Other determinants of health are considered. In the absence of clinical data that can comprehensively and reliably describe the population of northern BC—including distinct First Nations communities—health professionals’ understanding of local health issues and delivery of appropriate healthcare solutions must rely on generalized research, guidelines, and standards of care derived elsewhere.

Contemporary health and healthcare delivery are increasingly indexed to predictive, preventive, and participatory (P4) models that require an understanding of both the molecular and environmental basis of human disease and the need for personalized care [5]. While P4 models are increasingly accepted, they require a shift towards consideration of physical, emotional, mental, spiritual, and social facets of health and wellbeing through culturally safe healing approaches [6,7]. The healthcare system could facilitate this balance by striving to achieve equity via a “universal” healthcare system in Canada [3,8], as molecular data continues to offer key insights for the prevention, diagnosis, and treatment of disease. However, progress in understanding health and disease requires an equitable opportunity to participate in research, including biobanking and affiliated genomic research. Without equitable access to research, inclusion and personalized approaches to healthcare delivery will continue to be limited. Indeed, as progress in these fields grows among the general population, the subsequent disparities will not only persist but also expand.

An increase in biobanking, as indicated by the increase in publications in recent years, has enabled growth in the research platforms necessary to support genomic research [9]. This has facilitated rapid evolution in the capacity of precision medicine and advanced innovations for addressing chronic conditions [4,10]. In precision medicine, individual patients are prescribed safer, more effective medical care tailored to their specific needs [4,11]. While the technologies and developments are innovative, individuals living in metropolitan areas or receiving care in academic healthcare centers with higher socioeconomic status stand to benefit disproportionately from genomic research in comparison to those living in northern, rural, and remote communities without easy access to research facilities, participation, or specialized care [12]. Most Canadian biobanks are based in southern, metropolitan areas [13], which underpins this imbalance. Indigenous Peoples in Canada (collectively First Nations, Métis, and Inuit) remain particularly disadvantaged and under-represented, falling into the globally acknowledged and widening “genomic divide” [14,15,16,17]. This has implications for future genomic health research and the potential healthcare benefits of translational “bench-to-bedside” or “bedside-to-bench and back again” approaches with Indigenous populations [2,18].

There are three distinct Indigenous groups in Canada: Inuit, Metis, and First Nations. There are over 600 recognized First Nations bands in Canada, representing a diverse array of unique cultures, histories, and traditions, including language, ceremonies, and healing practices. Indigenous Peoples in Canada continue to endure intergenerational effects from colonialism, experience gaps in socioeconomic and health status, and have decreased access to health research in comparison with non-Indigenous Canadians [19]. Further, colonialism and associated government policies have involved well-documented, egregious ethical transgressions in health research involving Indigenous Peoples in Canada, the United States, and other global jurisdictions [20,21,22]. Colonialism continues to manifest itself in structural barriers and culturally unsafe outcomes when it comes to health research and Indigenous communities. As a result, genomic research elicits serious and valid concern amongst Indigenous communities, as demonstrated by our findings. This lack of trust stems from current and past abuse by researchers and the healthcare system in advancing priorities that are misaligned with community desires or are malicious and detrimental to communities and rights to self-determination [23,24]. The resulting inequities, marginalization, and mistrust continue to fuel the “genomic divide” specific to Indigenous Peoples, resulting in truncated access to the potential benefits that innovative, contemporary research could provide [17]. In BC, there are 203 recognized First Nations communities, the most of any province in the country. First Nations in BC also have the greatest diversity, with 26 cultural groups and 34 languages (more than 60% of the country’s First Nations languages). In total, 155,000 (3.6% of the BC population) people in BC self-identify as First Nations, the second highest population amongst Canadian provinces and territories [25].

In BC, long-standing gaps in health exist between residents in northern BC and those living in the south, and between First Nations Peoples and the rest of the province [26,27]. Complex, multifaceted determinants hinder access to healthcare, biobanking, and genomic research for northern BC’s First Nations communities. Healthcare access issues include transportation challenges, limited financial resources, meager or non-existent accommodation options, inadequate support for travel companions, transient healthcare workers, and a lack of childcare options, to name only a few. These greatly restrict patient access to primary care, specialist consultations, diagnostic tests, and procedures [28,29]. Determinants driving inequities in access to healthcare are translated into mounting gaps in access to research, with additional challenges including large geographic distances from our metropolitan-based academic healthcare facilities and the affiliated research and a paucity of Indigenous healthcare providers (HCPs), academic researchers, and genomic scientists. In addition, there is a persistent lack of Indigenous voices at the table, which drives research initiatives, funding, and decision processes [2,30,31,32].

As health research continues to facilitate cutting-edge innovations globally, current health discourse emphasizes the importance of addressing these health gaps. However, pursuing this agenda on a large, national, or international scale can inadvertently relegate Indigenous Peoples to an agenda of persistent—or expanding—disparity [33]. Thus, more local approaches are advised. Addressing these gaps requires bringing novel technologies and research platforms to the fore by engaging and continually consulting Indigenous Peoples in respectful, collaborative relationships around the processes of research. This approach enables an Indigenous-led research agenda for the benefit of one’s own community [1,20,34]. The development of research capacity in northern BC and the creation of the northern biobank aim to address such knowledge and research gaps.

The Northern Biobank Initiative (NBI), which commenced in 2012, is a 6-phase research project (see Table 1) that aims to collaboratively engage 54 First Nations in northern BC in the co-development of a First Nations biobank in this region. This paper focuses on Phase II (NBI-II), the planning phase, which was commended in 2016, following four years of extensive preparatory dialogue about biobanking between researchers and First Nations community partners and stakeholders. NBI-II was the formal consultation with members of 54 First Nations around the governance, access, consent procedures, cultural safety, and protocols of the proposed establishment of a Northern BC First Nations Biobank (NBCFNB). This paper describes the process and results of this large multi-community consultation.

## 2. Materials and Methods

### 2.1. Study Design

The aim of the NBI is the co-creation and deployment of a population-based biobank with clinical data and tissue samples from people in northern BC, with the potential to create an embedded Northern BC First Nations biobank (NBCFNB). NBI-II is the second phase of the NBI, called the planning phase, in which the NBI research team aimed to undertake a large multi-community consultation in northern BC to establish First Nations communities’ views and perspectives around biobanking to guide and inform the potential establishment of the NBCFNB. Specifically, (1) consultation was undertaken to explore and garner community views about governance, access, consent procedures, cultural safety, and protocols for the potential NBCFNB; and (2) the creation of a Northern First Nations Biobank Advisory Committee (NFNBAC) to be an integral part of the process to establish the NBCFNB. The consultation was conducted through a mix of key informant interviews and community member focus groups.

### 2.2. Research Setting and Participants

The NBI is situated in the Northern Health (NH) region of BC, Canada. This region, with a large geographical footprint (~600,000 km^2^) [35], has a population of ~300,000 people [36], with almost 18% identifying as Indigenous [37]. There are 54 First Nations within the NH region, with the majority situated in rural and remote localities. In addition to NH, this region’s healthcare is serviced by the First Nations Health Authority (FNHA), a province-wide authority that supports First Nations Peoples with health and wellness resources [38]. The physical location of the proposed biobank(s) (both the general NBI for northern BC participants and the potential NBCFNB) will be the University Hospital of Northern BC (UHNBC) in Prince George, BC, on the traditional, unceded territory of the Lheidli T’enneh First Nations Peoples. Consultation activities with First Nations communities occurred within the three geographic sub-regions [Northwest (NW), Northeast (NE), and North Central (NC)] in northern BC, as delineated by the governance structure within the FNHA Northern Region. The NBI is a vital platform to strive for equitable choice for participation in translational and clinical genomic research that promotes P4 approaches sensitive to the unique and diverse cultures of NH.

NBI-I involved an initial dialogue with health leadership, including the FNHA, BC Cancer (BCC), the BCC Michael Smith Genome Sciences Centre, Genome BC, the Provincial Health Services Authority (PHSA), NH, researchers, family practitioners, and First Nations elected political leadership (often called Chiefs), as well as health directors (Among the 54 First Nations in northern BC, there is some variation among governance formats for elected political leadership (Chief or Deputy Chief, Spokesperson, Mayor, etc.) as well as in health service delivery to community members. In some cases, the role of “Health Director” may carry a different title, e.g., health manager/lead, while upholding a similar job description and set of responsibilities and community members. Through the resulting partnership, the concept of the NBI and its methods and knowledge translation strategies were endorsed, with input from leaders of the Canadian Tumour Repository Network, University of BC (UBC) Office of Biobank Education and Research, and other biobanks in Canada. Subsequently, NBI-II was funded through the Genome BC (GBC) and four partners who were invested in its rationale and committed to its processes: FNHA, NH, BCC, and PHSA. Research participants were adult members of 54 First Nations in northern BC. Two groups of participants were sought: (1) Key informants: northern BC First Nations Chiefs, Health Directors, First Nations Council members, and Community Health Centre staff; and (2) focus group participants: First Nations community members.

### 2.3. Community and Ethical Approvals

In October 2016, after four years of dialogue and presentations, First Nations Chiefs and elected leaders at an FNHA Northern Regional Caucus meeting passed a resolution to support the establishment in principle of an NBCFNB in partnership with the NBI team. This resolution allowed for: (1) the NBI team to consult with First Nations in northern BC to receive input on the governance, access, consent procedures, cultural safety, and protocols for the NBCFNB; and (2) the creation of a Northern First Nations Biobank Advisory Committee (NFNBAC) to be an integral part of this process. This commenced upon Research Ethics Board (REB) approval with the University of Northern BC (UNBC) harmonized with UBC’s REB. Embedded in these REB applications was the commitment to follow FNHA governance processes and approvals prior to initiating contact with potential participants. It was critical that both FNHA and NH were active partners in consulting about (and ultimately developing and implementing) the resulting NBI and potentially embedded NBCFNB. Unique Indigenous First Nations governance considerations for this potential First Nations biobank are critical and will bring greater understanding to true partnerships in research initiatives between First Nations governance (FNHA) and Canada’s otherwise provincial- or geographic-based health authorities.

### 2.4. First Nations Governance

Initial steps of developing the project’s governance structure were guided by partnerships with regional First Nations communities, key health and research stakeholders, and supported by international and local allies, the FNHA and NH. The governance and project scope were structured and sustained through consistent engagement with the NFNBAC and FNHA Northern regional leadership. The next steps needed for the creation, implementation, and operationalization of the NBCFNB will be the consideration of the roles of NBCFNB governance body members and approval and establishment of the proposed governance model that has emerged from these years of consultation and partnership building in Phase II that this paper will describe. Project ethics and protocols are guided by OCAP^®^ (OCAP^®^ is an expression of First Nations jurisdiction over information about their communities and their members). As such, OCAP^®^ operates as a set of specifically First Nations—not Indigenous—principles. https://fnigc.ca/ocap-training/ (accessed on 31 January 2023) principles, where First Nations Peoples have the right to Ownership, Control, Access, and Possession of their data and information, regardless of where that information is held or stewarded [30].

### 2.5. Data Collection

Methodology, subsequent consultations, and knowledge translation strategies were planned and conducted with input and guidance from the FNHA Northern regional leadership, community input via FNHA caucus presentations and discussions, and formal advisement from the NFNBAC.

A qualitative approach suited this exploratory process, as the interface of genomic medicine and Indigenous self-determination continues to receive little attention, and as such, an exploration of firsthand knowledge from engaged participants into what affects them in this field was long overdue. This qualitative inquiry emphasizes: (1) The understanding of the consultation and partnership-building processes with First Nations Peoples and the reasons they unfold as they do; and (2) the importance of conducting research *with* those most affected by the research topic. This ensures that the proposed study, the emerging NBI, and the potential NBCFNB are based on information, perspectives, and recommendations from First Nations Peoples affected by the inequity in healthcare services and research in this field.

Recruitment approaches differed for key informants (KIs) and focus group (FG) participants.

KIs (elected leaders or proxies and health leaders or designates) received formal invitation letters by email after the process was discussed at the Northern Regional Caucus meeting with FNHA leadership and the NFNBAC. Interviews were then arranged using a mix of follow-up e-mails, phone calls, or by approaching potential KIs at FNHA northern BC sub-regional and regional caucus meetings to allow for broad representation. Before any KII questions were asked, briefing notes were given to KIs and the interview allowed ample time for information sharing to address any questions the KIs had about the NBI, the interview, or their consent. Given formal positions of KIs, most had attended one or more NBI presentation(s) at FNHA events provincially, regionally, and/or sub-regionally prior to this KII process. KIs were then individually consented, availed of confidentiality agreements (online, via an audio recorder, or using paper consent forms at the onset of in-person KIIs), and offered an interview choice (phone, online, or in person at their preferred meeting place), allowing privacy for formal and open communication. Interviews were semi-structured and approximately 45 min in duration.

The selection of FG participants was semi-active. The NFNBAC and FNHA Northern regional team leads recommended and approved host communities and provided access to Community Engagement Coordinators (CECs) who then worked with Health Directors and local health staff to identify potential FG participants—those considered by fellow community members to be a “voice” for the community and willing and able to contribute during a 3 h long FG. Such recommendations were not based on previously known perspectives on this topic and were considered representative of the community. A mix of genders, ages, and occupations was represented in the adult population, including Elders and Knowledge Keepers (Elders are respected older adults who are held in high esteem by their communities for their wisdom, cultural, and traditional knowledge. Knowledge Keeper is a term used to encompass a variety of holders of traditional knowledge, including but not limited to language, culture, spiritual practice and wisdom, traditional science and skills, arts, medicine, harvesting, hunting, and land guardianship, to name a few). CECs provided potential participants with formal community member FG invitation letters. Those who wished to participate signed informed consent forms and individual confidentiality agreements and then registered to attend an NBI information session on the basics of biobanking and examples of applications of genomic health research. To optimize the level of discussion in FG, these sessions were required for FG participants but were open to all community members. The FG scripts were semi-structured questions reviewed and approved by the NFNBAC. FG participants were asked to rate their own knowledge of biobanking and genomic research, before and after attending the NBI information session, and after participating in the 3 h long FG discussion. As per the guidance of the NFNBAC, each FG included a local Elder/Knowledge Keeper who opened and closed the FG event in a culturally relevant way based on the traditional lands upon which the FG occurred. This was intended to provide a level of cultural safety, ensuring that FGs proceeded in a good way. In line with FNHA protocols, KIs and FG participants were offered culturally relevant gifts in appreciation of their time and contributions to the NBI consultations. FG participants were provided honoraria and reimbursed for travel and accommodation costs for participating, as the CECs aimed to diversify input as much as possible across a range of surrounding First Nations communities. Standard FNHA protocols for such reimbursements were followed.

### 2.6. Data Analysis

KII and FG dialogues were recorded on digital voice recorders and transcribed verbatim in Microsoft Word using ExpressScribe software v7.03. The analysis of KII and FG data was informed by and structured around the project’s objectives of: (1) increasing awareness of the NBI, biobanking, and genomic research among First Nations community members; and (2) identifying First Nations Peoples’ understanding and experience of biobanking, perceptions of key issues, challenges associated with biobanking, and recommendations or expectations for establishing governance for the NBCFNB in a beneficial and culturally safe way. Descriptive analysis and analysis of variance (ANOVA) were performed in Microsoft Excel, including a one-way between-subjects ANOVA to measure the educational effect (via self-rated knowledge) of attending the NBI information session and subsequent FG. A qualitative analysis of the KII and FG transcripts was conducted using NVIVO11 [39] software. One researcher (KA) reviewed the transcripts line-by-line using an interpretative phenomenological approach to develop a thematic framework [40] reflecting FG participants’ views about the proposed creation of an NBCFNB. The details of the qualitative findings were written up into a community-friendly report, which was provided to the FNHA Northern Regional Table, northern BC First Nations community leadership, and representatives at several key gatherings in 2022 and is presented here.

## 3. Results

### 3.1. Participant Demographics

Thirty-two KIs were interviewed, of whom 29% were elected Chiefs of northern BC First Nations, 42% were health directors, and 29% were First Nations Council members or staff from community health centers, such as community health nurses or community health representatives. The representation of KIIs and FGs by sub-region was relatively equal based on population (Figure 1 and Figure 2, respectively).

### 3.2. Community Member Self-Rated Knowledge of Biobanking and Associated Genomic Research

Across all FGs, community member self-rated knowledge showed a statistically significant increase after both the NBI information sessions and the FGs (Figure 3). There was a significant increase in self-rated knowledge after the information session [F(1, 88) = 33.51, *p* < 0.0001] with another significant increase in self-rated knowledge on similar parameters before and after participating in an FG discussion [F(1, 87) = 7.08, *p* = 0.009].

### 3.3. Response Frequencies for KIs and Community Members in FGs

KIs’ responses revealed many challenges to the participation of First Nations Peoples in biobanking and genomic research, but also “good way” approaches that could be remedial (Figure 4A,B).

Challenges identified included: lack of trust and/or prevailing mistrust from past/present abuses in healthcare and research; lack of awareness around biobanking/research in general; ongoing barriers to accessing or inaccessible healthcare; uncertainty concerning research and associated benefits; lack of language translation/non-medical communication support for community members (especially Elders/Knowledge Keepers); rushed “informed” consent processes; and other barriers pursuant to financial limitations and low socioeconomic status. Remedial “good way” approaches suggested by KIs included: promoting better communication (including direction, intention, and benefits involved); engaging First Nations leadership early to gain support; building relationships of trust; promoting cultural safety throughout the process; supporting Elders; and building a First Nations governance model.

Community members taking part in FGs echoed similar views (Figure 5), with the building of trust in the process and control over the governance structure for the proposed NBCFNB being the most highlighted anticipated challenges.

#### Remedial Approaches in a Good Way

The presence of trust and control would empower First Nations to feel confident in participating in their NBCFNB. Improving communication and awareness about their NBCFNB and its intentions and benefits to First Nations were the most important “good way” approaches in the estimation of both KIIs and FGs. Offering resources with visuals and simple terminology was expressed as important to ensuring understanding and engagement. Cultural safety and consent were common expectations voiced by community members involved in FGs.

### 3.4. Qualitative Findings

Qualitative analysis revealed nine broad themes regarding the concept of an NBCFNB. These themes and subthemes are illustrated in Figure 6 and described in further detail below.

Empowerment through communication and knowledge sharing

Understanding and informed decision making

FG participants expressed varying degrees of uncertainty about the nature and aims of a biobank. While all participants had attended information sessions on the NBCFNB, many reported that their understanding of biobanking was insufficient to make specific or detailed comments on the development of the NBCFNB. Several participants outlined the need to understand the NBCFNB and the scope of possibilities in more detail. It was generally expressed that the round of consultation being undertaken was a critical first step in building awareness and understanding and empowering First Nations Peoples to make informed decisions around support for and involvement in an NBCFNB.

Note: In all instances, quotes are clean and verbatim, edited only to remove filler words (such as “um”, “uh”, “like”, “you know”, etc.). Quotes were not changed in meaning. Any additions for clarity are reflected in [square brackets]. When quotations have been shortened, this is indicated by ellipses (…). Full-verbatim quotes with filler words intact are available upon request.


*And so, the process shouldn’t be just about the gold statement or business statement or mission statement, it needs to go back to the community where the people can interpret, allow them to interpret also what it means to them, what biobank means to them, and how it means to them as First Nations people.*
(FG4-P5)


*The communication is really, really, key like we were talking about language, and just getting people to the point where they can see the benefits, right? But this isn’t something that happens in a year, we’re talking decades. Right, this is ongoing, it’s transforming, it’s growing. But I think that there’s much to be gained from this.*
(FG2-P3)

bMeans of increasing understanding

Participants emphasized that it would take time to build the requisite knowledge in First Nations communities about the benefits of biobanking, in order to foster strong partnerships and uptake. Decision making around involvement in biobanking was stressed, considering past traumas associated with research. Methods suggested for ensuring effective knowledge sharing and decision making around the development and operation of an NBCFNB included: training local community champions to promote and educate communities; developing “cheat sheets” suggesting key terms to aid discussions with health professionals around biobanking; developing and distributing visual communication resources; ensuring that functional descriptions of key concepts are developed and utilized; and using the familiar repetition and ongoing dialogue that defined NBI-II. The need for non-scientific language in this ongoing dialogue, creation, and implementation of the NBCFNB was repeatedly voiced.


*Creating champions within the community and having contact people within the community that have the proper information of the benefits and the risks of this. Just sharing that information within the community, having information sessions, really having a promotion campaign about it.And incorporating traditional ways of gathering and how we share our information within each community, identifying what’s unique for each community, and the biggest thing is sharing the information.*
(FG5-P4)


*… it’s key that they are informed, very well informed. You would need to have a translator to speak to what the work that you are doing in the language because ‘what do you mean by DNA? What do you mean by northern biobank? You know, exactly what does that mean?’ And that needs to be interpreted for them. And not just the elderly but also to the to the younger generation next becausenot everybody’s well-versed in this biobank, well actually nobody is.*
(FG4-P5)

A website for the NBCFNB, social media communications, diagrams, pamphlets, and videos about biobanking shown in doctors’ waiting rooms were suggested to increase community knowledge about biobanking. Such resources would need to be adapted to meet each community’s needs across northern BC.


*Go to assemblies or any gathering, health gathering, or even go to the band council meetings of each community and have a presentation and booklets, it’s good to have a presentation but leave some information behind… and I think one of the things that you know young people…they are good with computers. If we do have a website certainly have that available.*
(K106)


*I really appreciated that video *[that was]* played at the information session…a lot of people may not be able to grasp that [referring to biobanking] just by talking to somebody. A lot of people are visual learners so if there’s something that’s already out there that kind of explains that. It might help. And then also having, well it would differ from nation to nation, having it explained in our language.*
(K128)

While social media was suggested, some participants were wary of its use due to the potential for sharing information with people outside of the community in the case of personal opinions and perspectives shared for community purposes only.


*Social media works but it’s a touchy subject because we get worried about outsiders knowing.*
(FG7-P2)

Effective and appropriate communication around all aspects of NBCFNB operations was identified as important to meet different community and health needs. Such communication includes attention to power imbalances, jargon and medical terminology, health literacy, and the timing of sensitive discussions. Ensuring people are afforded sufficient space, time, and information to understand and consider involvement in a biobank, free of pressure to be involved, was identified as being fundamental.


*Because in First Nations culture, they don’t rush. Things are not rushed. The pace is very different. So I really think this work is really important and I think if the community can understand what the goal is…I think they’re really going to want to volunteer to do and promote this research.*
(K123)

2Establishing and Rebuilding trust in research

Context and cultural safety as conditions of trust

Establishing and rebuilding trust emerged as a key priority to address the trauma and resultant mistrust caused by the ongoing effects of colonialism on the experiences of many First Nations Peoples in healthcare and affiliated health research. Ensuring that all aspects of an NBCFNB are designed to recognize and respect this history of trauma was suggested as a necessary foundation for moving forward. Some participants spoke of brutal and denigrating studies conducted on First Nations Peoples without their knowledge as a major impediment that an NBCFNB would need to overcome. The most common examples were the scientific studies conducted in government-funded institutions in the era of Indian residential schools and Indian hospitals [41,42]. Building upon current knowledge about biobanking and its potential benefits for First Nations Peoples was seen as important in building and regaining trust and enabling First Nations’ control and ownership of the biobank subsequently used for research. A true partnership is expected.


*I think other benefits is to ensure that it is culturally safe and to ensure that it’s done in a way that our people can be proud of and feel safe to participate in. And that there is a cultural and traditional aspect to the entire project. I think the only risk or even apprehension that I can foresee is that First Nations people were heavily researched. And not with consent. Especially during residential school, the numerous experiments that were done on our children. And so it could hit a nerve in some communities. So, that’s why I feel it’s very important that it [the biobank] is rolled out in a very respectful and culturally safe way.*
(FG1-P3)


*The whole way that community or the clients that you might encounter or people that you wanna have participate in this research, the trust factor is huge, because of the trauma response that people seem to have. Avoidance is, really deep, right?*
(KI23)

bControl and commitment to relevant research

Skepticism and wariness were voiced about who is controlling the research agenda and outputs. Suspicions were communicated about the government and/or resource companies wanting to collect and use genetic information from First Nations Peoples to build cases to undermine First Nations land claims and/or justify mining operations (a prominent economic resource extraction industry that impacts many First Nations in northern BC) while avoiding litigation over the negative health impacts of the industry. Any partnership with industry, pharmaceutical or other, would require First Nations governance control.


*I’m on the fence on this whole thing. Becausethere’s always new ideas coming along and there’s always people coming to consult with us, First Nations, and on behalf of us kind of thing… I’m not questioning the integrity of your program, it’s just general…we’ve heard it a million times before, right, not just from you guys but every oil company, every government, saying the same old kinda thing, right?*
(FG3-P4)

3Ensuring that research benefits First Nations Peoples

Evaluating the risks and benefits of research

Concerns regarding whether and how research from an NBCFNB would benefit First Nations Peoples were commonly expressed. Concerns were voiced around the possibility of research being undertaken using biobank material that promotes negative stereotyping and/or victim-blaming of First Nations Peoples. The expectation that an NBCFNB would work in partnership with First Nations communities to ensure all research undertaken on biobank material would benefit and not disadvantage First Nations Peoples was commonly expressed. Establishing open communication was cited as critical in planning structure, governance, protocols, and scope of research within the NBCFNB, particularly regarding communicating the realistic benefits of research to the community.


*The best communication for this is to have regular community meetings to explain exactly what biobanking is, and what’s the benefit, and if you’re gonna… Like, I’ve heard you say that it’s gonna be focused on cancer. Okay, then, you have the experts that are doing this … Even if it’s the final product to make the community more aware of what it is, what it entails, and how it’s gonna benefit our community, and to ensure the community understand that this is not a cure, it’s more information, and hopefully, a potential option that may arise is that it could lead to maybe intervention, prevention.*
(KI26)


*So, you see that, whenever you are talking to the traditional healers and so on, they are very guarded because they do not want to give away their knowledge of plant medicine, which some corporate entity from far behind pulling the strings will be able to determine and go and take and make use of it without anything coming back to them.*
(KI22)

bCommunity-driven research

Prioritizing issues of interest or concern to communities in the research undertaken by an NBCFNB was important, with participants expressing that biobank data should be available to answer questions First Nations Peoples have about illness and disease. Some health areas of interest included cancer, the health impacts of externally driven environmental changes, such as polluted drinking water, dams, changing weather patterns, toxins in food sources, inability to hunt or gather traditional foods, and proximity to mining-related activities such as fracking, and how these relate to the increasing rates of cancer and other seemingly increased risks of chronic illnesses among First Nations Peoples. Other participants voiced interest in creating community-level health status baselines to measure the effects of environmental changes on the health of future First Nations generations. Participants were made aware that the proposed NBCFNB would not include all the necessary elements to commence such research. However, in the context of cancer (which was a prominent topic in FGs and KIIs), the NBCFNB could be a research tool to enable participation in research aiming to provide a genomic-level understanding of tumors.


*I think it’s extremely important with the amount of industry and natural resources, especially up here in the north. There’s a lot of industries that are coming in and it’s, I think there’s a lot of effects on the natural and traditional foods and the impact, although there’s not a whole heck of a lot we can do about it, but at least having that information that there is an impact. And I’m astonished at the amount of cancer rates in the XXX [named community]. It seems like they’re exploding exponentially, so having a biobank would be really important to see if there’s some linkages or some risk factors or something that we’re missing that we could actually start addressing for future generations.*
(K120)

cTraditional knowledge

The acknowledgement, respect, and incorporation of traditional knowledge in the operations of an NBCFNB were seen as particularly relevant. Incorporation of First Nations Elders and Knowledge Keepers in planning and development from the biobank’s inception was recommended to advise on cultural protocols and relay communications to First Nations Peoples on the nature and value of biobanking.


*The other piece I wanted to add too that we do need to have somebody with traditionalhealing or traditional practices. ‘Cause sometimes with those samples there may be a ceremony, there may be taboo, there may be something around samples and body samples, so just recognizing that there may be traditional things that have to happen for those processes.*
(KI20)


*I’m wondering if there should be some form of ritual added in, whether if it’s a giving of tobacco for taking the sample or whether it’s outside or a prayer or a sage… just an honoring of the spirit of what it’s going to…*
(KI14)


*I know that there’s a lot of spiritual reasoning behind people not wanting sampling done or that sort of thing, you know, any tissue samples taken for that purpose. So find a way to, a practice or something, go to the Chiefs maybe, and see what their suggestions are, or any First Nations leaders within the community.*
(KI09)

It was suggested that traditional knowledge about disease and treatments be explored by considering the collection of data from First Nations’ perspectives. It was proposed that the NBCFNB, in true partnership with First Nations Peoples, would enable any health or economic benefits from traditional knowledge or medicines to be realized by First Nations communities and not exploited for the benefit of others. The clinical and personal data to be collected for the NBCFNB will ultimately be determined through an established First Nations governance structure to address this sensitive issue robustly.


*And the bank…that’s a lot of information that can end up being used in so many great ways. And like I said before with our natural medicines, because there is no research, if somebody chooses, you know, ‘Well, hey, I’m not going to go with this chemotherapy or I’m not going to go with this, I’m gonna, you know…’ then we have something that’s going to be for future that’s like, ‘Yeah, our medicines do work.’ Then you got that side, for, ‘Uh-ho! All our medicines work!’*
(FG2-P6)


*Yeah are we curing cancer, or you know if, are we gonna create something, are we gonna go into traditional medicine?*
(FG2-P5)


*I would suggest when that tissue is taken, ask whether there is natural medicine used or not. And what was actually being used, because that could actually alter the samples. Like was soap berries used, was spruce used, was chaga used? Which isn’t a normal question either. But some of them may have used it and maybe it’s altered it or shrunk it, that would all be viable information because looking at some of the traditional, did it affect that cancer? Has it changed it, did it shrink it, is it an aggressive cancer that didn’t actually kill because it was treated properly?*
(KI14)

dData sovereignty

Many participants were aware that there are examples—in Canada and abroad—of research being conducted without communication about research outcomes or implications. Participants described traditional healers as being unwilling to share knowledge with researchers due to past experiences of cultural misappropriation and monetary exploitation. The prioritization of data sovereignty and biobank access protocols, considering local, community-based, and leadership priorities, was described as essential to involving communities as active research partners. This would further enable communities to exhibit ownership over data and respond proactively to concerns within their own communities. The OCAP^®^ principles were further referenced as important guides to NBCFNB’s processes of data collection and governance [43]. This dialogue was balanced with ongoing questions about a collectively established governance structure that these consultations also explored.


*Well, the more information that we have you know, the better the chances, but also we just have to be careful because this information can be used by some, some ways that other people may use it as monetary gain or whatever or other ways that can benefit others. Not in a way we are thinking of. But mainly I think it’s important the more information we have the better.*
(KI06)

4Fostering respectful community engagement

Community engagement and engendering trust

Given the context of wariness and mistrust concerning research among many in First Nations communities, taking time to engage with communities about risks and benefits was regarded as fundamental to ensuring that First Nations Peoples felt safe to support and participate in the NBCFNB.


*I think what you’re building here by coming to us, explaining it the way you are brings in the trust for me. I don’t know how everybody else feels but I think that’s huge for us. We’ve had so many other things done to us and so thatwe don’t trust too many people. But I feel the trust coming in by you explaining it so nicely and talking to us so we understand what you’re talking about.*
(FG6-P2)

Participants often explained that they were not able to speak on behalf of entire communities and could only convey their own perspectives, affirming that decisions around biobanking participation should be made individually. Some suggested that communities should be broadly engaged in decisions around supporting an NBCFNB. Recommendations for ongoing community-level consultation and engagement included: hosting community dinners with a culturally appropriate video or presentation; developing questionnaires for Elders; and assisting community champions to engage within their communities about biobanking. Participants expressed dissatisfaction with past governmental attempts to engage communities via means such as letters and postal surveys, with a preference for face-to-face engagement emphasized (while it should be noted that these formal consultations were completed just months prior to the COVID-19 pandemic being declared by the World Health Organization in March 2020). Pictures, diagrams, and videos were often given as options to accompany written documentation and surveys.


*When we normally do our information sharing to communities, if we do it the Western way, we’d be lucky if we get half a dozen to a dozen people in attendance, but if we do our within our culture, which is the feasts, some people call it the potlatch, we’ll get anywhere from 75 to 250 people.*
(FG4-P2)


*If you could give us a questionnaire paper, you could bring to the [reserve], and just straight-out basic questions and then we could go around and do a survey to the Elders and those people in general and see what kind of feedback we get. And just see how they feel about it.*
(FG5-P1)

5Building research capacity among First Nations to address their priorities

Who decides and what is decided upon?

Significant interest was conveyed concerning the nature of research that would and could be conducted by an NBCFNB, including discussion around *who* would be conducting the research and operating the biobank and *what* would be investigated using the biobank platform.


*I feel that it’s important for us to research that and have control over it ourselves, so we know it’s being taken care of sacredly. Each one of us have a spirit and a soul. And for mine to be out there is a big risk to allow people to research me for whatever reason there is. But I also feel it’s important the name attached to it, and I pray that this doesn’t break down in the future. Because it’s so important to have this. For my generations to come.*
(FG7-P3)

Participants emphasized that efforts be prioritized to empower communities as active agents in deciding *who* would be conducting NBCFNB-associated research. Training First Nations Peoples, especially young persons, in research skills and therefore building capacity was considered positive in building trust around research and ensuring its conduct in culturally appropriate ways. Similarly, employing researchers to conduct studies was deemed an important element in establishing trust with communities.


*If we had more First Nations people actual in the, the process. I mean I think it’s a good idea generally, but just in a First Nations context, because we’ve been just [wrung] with the coals how many times right? And someone’s coming in and is like ‘oh we’re gonna help you’ and then, you know, then they get what they want and then they’re out of here.*
(FG3-P4)

It was suggested that processes be established for communities to be presented with the findings of any research undertaken with their members’ data, so the community could determine and control whether the findings were suitable for release and publication. There was some doubt expressed about the willingness of healthcare professionals to relinquish this measure of control and how a similar challenge should be mitigated in the research arena.


*It would be interesting to see too the transition of information between First Nations people and the medical professionals because right now it’s the family physicians that are in charge of our files. They know everything about us. … And so, when we meet with them, we talk about the database, that it would be good for us and a big question is ‘are they prepared to share that information with us? Are they open to us having access to their data?’*
(FG4-P5)

6Incorporating cultural safety into the NBCFNB process

Embedding Culture into the NBCFNB

To ensure a welcoming and respectful environment, the embedding of diverse First Nations culture into the development and operation of an NBCFNB was commonly described as important. Incorporating First Nations protocols, ceremonies, and language into day-to-day NBCFNB operations was seen as critical to embedding First Nations identity and culture into the process and presenting a valuable model to inform change in other organizations servicing First Nations communities. The physical environment should be a relaxed space, including: choice of geographic location within northern BC for the NBCFNB; ensuring potential participants did not feel trapped or uncomfortable inside a big building when discussing participation in the NBI/NBCFNB; culturally appropriate artwork and furnishings; and friendly, welcoming staff trusted by First Nations community(ies).


*But you know, there’s a personal touch there, I think, I believe that makes that… says that, ‘Okay, um, it’s not so clinical,’ and I think if you’re going to have something set up for First Nations to come into, you know for a lab that’s traditionally Western, that would be really, really, clinical for a First Nations person.*
(FG4-P2)


*That might be where a traditional healer comes in too, to make a little ceremony to make it safe.*
(FG6-P10)

Communication relating to the biobank should be phrased in simple, non-medical, or non-scientific terms to ensure full community participation in dialogues and fully informed decision-making processes in the emerging governance structure. While it was often stated that engaging via face-to-face communication methods was preferable, it is recognized that these discussions were community-based prior to the COVID pandemic.


*…a lot of our people say that they don’t really understand the jargon of the professionals whether they be health or legal and so they like to narrow down the information and make it simple terms andnot too overwhelming, so I think it’s important to work with the staff andmake sure that they have a real understanding of it and then using them togenerate that information and flow it into the community.*
(KI19)

bEnsuring confidentiality

The issue of confidentiality protocols and governance was of critical interest and concern. Discussions centered around requiring the preservation of NBCFNB participants’ confidentiality when collecting and storing specimens and clinical data. Assuring that genetic information would be protected and secured in an NBCFNB, particularly from insurance agencies, was seen as important. Health research conducted in small communities has the potential to identify families or individuals at higher risk of developing certain diseases due to the overall low numbers of specific diagnoses involved. Potential stigmatization of individuals, families, and First Nations communities due to affiliated research was a prevalent concern. Accordingly, the requirement of the potential NBCFNB to critically assess risks to confidentiality, communicate with partners, participants, and communities about such risks and appropriate safeguards, and work to ameliorate these risks was a key priority.


*the end result […] there’s gonna be a paper that’s produced and it’s gonna have these First Nation community names on there […] And long-term it might not be a good outcome because the stigma that comes...obviously that’s gonna have some kind of impact socially between First Nations communities. I think that’s something that you might want to consider […] Cause your pool is not very big.*
(KI05)

cEnsuring transparency

The history of research conducted on First Nations Peoples and the resulting trauma and mistrust were highlighted in discussions around an NBCFNB. Concerns from these past histories revolved around examples of the attainment and use of tissue samples from Indigenous Peoples in ways that participants had not consented to or were not properly informed about. Clarity and transparency in all operations of an NBCFNB were emphasized in order to establish, re-establish, and sustain trust in research, including reporting the results of research undertaken with biobank material back to communities. Participants emphasized the importance of ensuring separation between governing bodies and the day-to-day operations of an NBCFNB, to reduce the risk of conflicting interests and ensure that governance can operate autonomously without the bias of researcher ambitions.


*I have a grandson, now. I don’t want that for him. So those things are all things that First Nations people have fears about too. Because, what will happen to the data? Can we trust people to do what they’re going to say to do, because we have this background where people didn’t do what they said they were going to do. Or they did things that were plainly unethical. You know, in the guise of science.*
(KI24)


*Yeah, and if there’s misinformation given, you can give that fear mongering, especially based on our history of people saddling us from the time. And sure, I have, I remember the doctors just doing really random crazy things and going ‘What did they do?’ … You know, so makes you question, so then I think there are always underlying factors of what information you have to give and what, how useful a biobank is for our people.*
(KI03)


*It’s distinct in thatthe board is somewhat overseeing and has that sense of responsibility regarding how this unfolds, or, operates and, however, in terms of the actual day to day in the operations and what occurs with the biobank, the board would not have direct contact with that.*
(KI21)

7Ensuring equitable research representation

Equitable access to research representation and health services was commonly reported as a critical issue facing First Nations communities generally in northern BC. A similar but more pronounced inequity was acknowledged to be access to biobanking and genomic research. Biobanks simply do not exist in First Nations communities, in northern BC, or in community hospitals serving rural communities in Canada [44]. Common barriers to access were complex and multifaceted, including issues of poverty, education, employment, isolation, geographic distance, meager transport options, poor weather conditions, childcare availability, cost, and trust. Concomitant factors such as the need for family members to accompany individuals, limited and expensive accommodation options to access health services and limited resources were described as compounding the ability of people to engage with a biobank.


*…there’s limited access, limited travel, you know, now that they’ve shut down the Greyhound bus (Greyhound is a public bus company no longer serving northern BC). We have that Northern bus and the Northern Health bus [provided by NH] but you know it’s hard to schedule things around those kinds of things, we have the trains but I think reliable transportation is a big piece.*
(KI25)


*So they don’t realize just how far people have to travel to go to get the help they need…Those are things that are really difficult for families. And when you get down there, having a place to stay. So economics, travel, isolation, lack of consultation.*
(KI17)

8Developing governance and protocols that support First Nations Peoples

A representative and multidisciplinary governing body

The need for a robust governing body to oversee the governance of an NBCFNB was a distinct focus of our consultations. Emerging suggestions for governance composition included diverse representation (age, sex, and gender) of northern First Nations communities from the three sub-regions as per the established NFNBAC and mirroring the infrastructure of the First Nations Health Authority Northern region [35]. Elders, Knowledge Keepers, and/or Traditional Healers were repeatedly suggested as vital sources and providers of cultural guidance and traditional wisdom for this governance body. Other areas of participation were broadly agreed upon (i.e., ethicists, lawyers, etc.), with efforts for those involved to be from northern First Nations communities, if possible. A draft governance structure was subsequently formed (Figure 7).

This type of multidisciplinary governing body was seen as central to (1) the facilitation of reciprocal learning for all involved with the biobank and (2) the sharing of information, understandings, and knowledge between the research world and First Nations communities. While having local representation from each community to facilitate active representation and communication about biobanking and help ameliorate fear or uncertainty about participating would be novel, the capacity for such commitment from every community was recognized as being too burdensome on communities. Active involvement in the well-established FNHA caucus system was repeatedly recommended, including presentations and written communication to these regular, formal gatherings of Chiefs, health directors, and others in the health and wellness portfolios of the 54 Nations in northern BC.


*So they’re also sharing what they’re learning with the communities so there’s less misunderstanding about stuff. So the more you bring in your cultural advisors, or just your Elder advisors that would play a huge role in opening it up to community. To see how what this works like, and how it, you know, and they can also guide you in that process in terms of cultural practices and cultural protocols.*
(KI08)


*The way that our Chiefs have organized themselves is there’s an appointed leader who consults with the rest of the chiefs of the Northwest sub-region and that’s our political rep for the Northwest. Not a leader, but a rep and so there’s one for the Northwest, one for the North Central, one for the Northeast.*
(FG4-P1)

Additional participants recommended that a governing board be comprised of people in a variety of roles, including: Indigenous partners; First Nations community members (including people with cancer or other health conditions); doctors and/or other health practitioners; researchers; and pathology or laboratory experts. Including an ethicist was considered particularly important for guidance around the complex and ethically charged use of samples in scientific research. Such ethical guidance must be informed by First Nations knowledge and ways of knowing to avoid further colonialism and subjugation of First Nations Peoples.


*So really making sure that that whole ethics and all that access is developed more from a First Nations point of view rather than a colonial point of view.*
(KI20)


*… you want someone who’s strong ethically, and like, community-wise and culturally, and we do have those people here, but how would you select? Like the North, even just the North is just huge. So do you wanta North Central person, a Northeast person, a Northwest person, or is it just one for the whole North? Good luck!*
(KI32)

Cultural safety training was suggested for all members and personnel associated with the biobank to ensure their preparedness for working respectfully with First Nations Peoples and establishing good relationships with communities and individuals. Participants noted that an NBCFNB built on this foundation would be a model for other organizations indexed to culturally safe service provision for First Nations Peoples.


*You’ll be the role model for the other organizations in the area with regard to any health service providers out there… Blazing the trail, is what the biobank can do actually with communication pathways between organizations.*
(FG4-P2)

9Ensuring that informed consent/declined participation is respected and aligns with established First Nations governance

The issue of “informed” consent

Numerous issues were identified to ensure that informed consent processes at an NBCFNB are robust and effective. Once again, the long history of poorly and unethically conducted research that failed to respect the rights of Indigenous Peoples emerged. Participants reflected on their own experiences of consent in the context of healthcare decisions, particularly around instances of miscommunication in cancer diagnoses, and strongly emphasized the importance of effective, appropriate communication in seeking consent from potential participants of the biobank.


*I know in our area, a lot of doctors and a lot of dentists, they tested on our people *[without their consent]*, … that could be a possible barrier, but also having thatsuspicion of, ‘What are you going to do with our DNA? What? Are you selling it or are you doing this…or whatever?’*
(KI03)


*I remember an incident where someone was diagnosed and didn’t understand the diagnosis and thought they were gonna die, and so you now got everybody in their family worked up because they didn’t understand that it wasn’t a malignant tumour, that they could have it removed and all of the process but they didn’t understand, they just heard cancer and everything else kind was muffled, so they just thought… ‘oh I’m gonna die,’ and started making plans for dying.*
(FG1-P3)

bEffective and appropriate communication

Ensuring that persons who chose to participate in an NBCFNB are accurately and appropriately informed was strongly called for, including recommendations for communication with Elders and members of First Nations leadership around consent to participate in a biobank. Suggestions included having a trusted individual, preferably First Nations and possibly someone from one of their communities, trained to oversee the consent process and ensure consent is informed, unbiased, and appropriate for different communities and age groups.


*I think that’s definitely something that, if somebody’s with a new initiative, any kind, that’s critical, but also continuing as you say that persistence, you have to maintain that conversation with Chiefs, Elders, and community members, and health leads, and so on.*
(KI09)


*So I think it’s really important to have First Nations leaderships or people who are in a leadership position to get their consent.*
(KI05)

The importance of location when communicating with potential biobank participants was often raised to ensure that they felt comfortable making decisions without pressure or coercion. This included the option of having conversations in home communities rather than city hospitals.


*I would look at another avenue to take to get that consent…and I don’t necessarily think it has to be in the hospital setting.*
(FG4-P3)

The need for respect and ceremonial acknowledgement and its importance to effective and appropriate communication was voiced.


*When we take something from a tree we acknowledge that tree, when we take something from a fish we put the bones and that back in the water. There’s things that are done because you acknowledge what you’ve taken. So do you acknowledge what you’ve taken from a person or is that a gift? It’s, I think you have to have that conversation.*
(KI24)

cWho should seek consent?

Some KIs suggested that doctors are best placed to seek consent to participate in a biobank. Other participants believe this should be undertaken by one or more experts through communication (explaining the nature and implications of participation in a biobank and appropriately seeking informed consent). The consensus around speaking with individuals about participating was that the conversations needed to be face-to-face, not rushed, and not place individuals or families under pressure. Face-to-face communication was considered a vital component. In such a large geographic space and after experience with virtual computer platforms during the COVID pandemic, this will likely need to be addressed in the future.


*I would really advocate, that as the biobank develops how this communication, and how this informed consent is going to be sought, you probably are going to need more than one person, because of the geography. And they’re going to need to be spending time and they’re going to need to be visible, kind of on an ongoing basis.*
(KI23)

dComprehensive or qualified consent

Questions were raised about whether consenting to participate in an NBCFNB would imply blanket consent for the inclusion of their material in all/any studies or whether consent should be selective and on a case-by-case basis. Some participants preferred that consent be sought for individual studies to improve autonomy over the use of samples. Others suggested that biobank participants could choose the types of studies for which tissue could be used at the time of consent. The implementation of such dynamic consent was flagged as logistically challenging but important to explore as it provides the highest levels of flexibility and control for those donating to the potential NBCFNB. The role of “permission to contact” participants in the future as an option in the consent form was discussed and generally approved as a topic to be discussed with the NBI and NBCFNB governance and ethics protocols. The emerging First Nations governance structure remained a key focus for the body that could potentially provide these decisions regarding access to the NBCFNB if the process was representative, deliberative, and transparent.


*That just because you give consent once, doesn’t mean that it’s a free for all and everybody gets to access it.*
(KI20)


*For me, one consent form doesn’t mean for everything, you know. I think every time there’s someone coming into the bank with their research, I want them to give me another consent form, you know, for anything and everything.*
(FG7-P1)


*It should spell out in the agreement with their donor [biobank participant] if he or she chooses to keep the sample in the biobank, it should say somewhat what they can do with it—they can say that do whatever you want with it or here’s specific things that you shouldn’t do. Give the authority to my daughter, son or whoever after.*
(KI06)

eTiming of seeking consent

Timing of the consent process was identified as complicated, since broaching the issue of biobanking with individuals or families dealing with newly diagnosed or palliative stages of disease would require great care and sensitivity. Preferred timing for raising the biobanking opportunity was often suggested to be while patients were undergoing active, longer-term cancer treatment, such as chemotherapy and radiation, or when they were told that such systemic treatment was not required. Participants believe this could help avoid some of the more acute and emotionally difficult times for cancer patients (i.e., the time of sharing the diagnosis, day of surgery, etc.) while recognizing there would likely be a range of opinions—and therefore potential options should be considered.

The timing of the first discussions with family about the sharing of tissue/data after the death of a loved one was identified as being challenging, particularly due to the different hereditary systems and lineage structures in northern BC communities. Including such issues in NBCFNB discussions and consulting communities to determine appropriate protocols for individual versus next of kin consent was supported. The NBI does not have current plans to pursue such posthumous consent, but this topic was brought forward and considered by participants in some FGs and KIIs.


*That one thing that our people really need to know is that they’re involved. If they’re not involved with any process of it then there is lack of interest in it. So if there’s a role that our leadership can play with any of the work that needs to take place in terms of advisory committee, then there’s more interest in it. ike with some community functions that I do, the hereditary chiefs are who we get involved becausewe operate through a cultural system.*
(KI08)


*I would say I would do at least one to two engagements about the project before you even sought consent […] Try to build the relationship and, and the word, get the messaging out as early as possible before you’re sort of going after the signature. Because that in of itself for some people is not something they want to do lightly.*
(KI23)


*… when you get a diagnosis of cancer, you don’t hear anything else. You don’t understand, half the time people don’t understand what’s being said. So when you present that consent, at that time or when you’re doing the initial surgery, you’re taking advantage of a vulnerable person.*
(KI24)

fPosthumous use of biobanked data and samples

The diversity of First Nations groups and different cultural practices was flagged as an important consideration in the posthumous use of biobanked samples and associated data to ensure respectful and responsive protocols and consent processes to meet individual and family needs. While differences exist, the commonly held belief that the physical body must transition “whole” from this world into the next was often identified. Thus, it was flagged that perhaps biobank participants should have the opportunity in the consent process to indicate that their samples be returned to family for burial or cremation upon their death. In these discussions, it was also acknowledged that such requests are not made for biospecimens taken purely for clinical purposes but are instead retained within the pathology department. Awareness of the great diversity in preferences and beliefs around the body and end of life was described as necessary for culturally sensitive engagement with potential biobank participants, a topic that will greatly benefit from input and direction from a First Nations-specific governing body.


*And then a few people around this area have come up and said ‘hey, our protocols are’—and the body being whole was actually specifically one of the ones mentioned to us, yeah, so that’s been really good.*
(KI09)

## 4. Discussion

The building of an NBCFNB will facilitate the creation of a research platform to enable First Nations to participate in genomic health research. Gaining access to the benefits of such research and personalized medicine in the prevention, diagnosis, and treatment of chronic disease has the potential to narrow the equity gap for the First Nations communities involved. The results of NBI consultations suggest that the information sessions and FG discussions were effective in meeting the first objective of consultations and community engagement—to increase awareness of the NBI, biobanking, and genomic research among First Nations community members. This is important as the NBI team and the NFNBAC hypothesized that increased health literacy would facilitate input and confidence among First Nations communities regarding participation in the FGs, in the NBI in general, and in interpreting the value and outcomes of genomic research. The mean score of 5.3 out of 10 is important to note as the increase in self-rated knowledge was a result of the NBI consultations, which for many marked the initiation of community engagement. Such engagement enables ongoing dialogue, raises awareness and literacy about biobanking and genomic research, and cultivates the relationship between the NBI team and community members. Increasing First Nations’ awareness and understanding of health and genomic research can empower confidence in generating research questions of importance to communities [45,46] and, as part of a multiphasic project, can represent a first step in sustained, collaborative “continuous conversations” with mutual respect [34].

The location of the biobank in northern BC is a considerable step towards achieving equity, serving as a platform to actively engage First Nations communities, perform health research that is responsive to their communities, and enable inclusive community-driven health research in their region. The First Nations consultations outlined in this manuscript demonstrated broad acceptance that access to high-quality, culturally safe biobanks is a significant step towards understanding health and disease in communities and closing prevalent health equity gaps. This initiative requires careful consideration of history and how it continues to affect First Nations’ well-being. Understanding the mediating factors of unbalanced power and socio-economics, self-determination, informed decision making, cultural safety, respect for privacy and confidentiality, Indigenous data sovereignty, and the cultural-ethical implications of building and maintaining this biobank are vital steps.

Establishing trust and First Nations Peoples’ confidence in, governance of, and participation in this potential NBCFNB depends on respectful, ongoing relationships. For many, this includes re-building trust in the concept of research, the Western education system, and clinical institutions where medical research is located. The process through which an NBCFNB responds to opportunities, reduces risks, and delivers benefits to communities in a culturally safe manner begins with respectful engagement that cultivates a strong sense of ownership of, and support for, the NBCFNB. This engagement is predicated on the Chiefs, leaders in First Nations health, and the appointed/elected NFNBAC members playing a key role in the inception, implementation, utilization, and maintenance of the NBCFNB. The intertwined participant optimism and concerns around an NBCFNB validate interest and investment in this project. Information from communities highlights the need for continuous conversations through a “living governance structure”. Ongoing communication will continue to inform the development, implementation, and operation of the NBCFNB.

Pursuant to the consultation and engagement process as outlined, we reflect on some of the main challenges, opportunities, and contexts associated with the establishment and management of an NBCFNB that were voiced by participants during the consultation process.

### 4.1. Establishing and/or Rebuilding Trust in Research

Mistrust and resultant hesitation of Indigenous Peoples regarding involvement in research is a well-documented concern [47,48,49,50,51]. Examples of previous exploitative research include that which: performed research without consent (i.e., ancestry); benefited mainstream science and/or medical industry without benefit to the community; was stigmatizing, discriminative, and disempowering; was incorrect or promoted negative data descriptions of Indigenous Peoples and their ways of knowing; and incorporated culturally insensitive research practices, including “helicopter research”.

This lack of trust in medical research is an ongoing concern that was also voiced during the extensive NBI consultations. The risk of an unfavorable cost–benefit balance was raised, but little was said about health research that worked well for the Indigenous communities involved. The journey towards reconciliation needs to be pursued steadfastly, allowing for the re-building of trust and promotion of self-determination. Key to this is appropriate protocols promoting effective and respectful engagement, ongoing and unrushed “dialogue to permit dialogue” with communities, and clarity and transparency in all deliberations [44]. It is important to note that genetic research with northern BC First Nations involving long Q-T syndrome was often referred to as a process to aim for and an outcome to obtain, led with the community by Dr. Laura Arbour, a medical geneticist and member of the NBCFNBAC [37].

Working towards enduring partnerships includes the promotion and incorporation of cultural safety and humility into the NBCFNB building framework. This could comprise the development of a value model to inform sustainable change in research with First Nations Peoples. Such a model could strengthen cultural partnerships through respect for and acknowledgement of community contributions in the NBCFNB implementation process. This can be actualized through the inclusion of First Nations members as valuable contributors or co-authors on manuscripts, publications, presentations, or culturally safe materials and resources about the NBI and the next steps towards this potential NBCFNB.

### 4.2. Building Research Capacity among First Nations to Address Their Priorities

Indigenous participation in biobank-related research and the wide range of associated benefits have thus far been limited [20]. To our knowledge, this is the first Indigenous biobank in Canada to be created for the purpose of future research (rather than genetic samples collected to address a known research question with secondary use of research data). Future research in this field at national and international levels is needed. The reasons for this limited participation are multifactorial and include: the northern, rural, and remote geographic distribution of many First Nations in BC; a shortage of trained healthcare practitioners in these communities; fewer Indigenous researchers and allies in these fields of research; and an inequitable distribution of resources and funding to support such participation [52], including the current absence of a First Nations-specific biobank created purely to enable participation in this research space. These factors impact communities that already lack the First Nations’ capacity to take on health-related challenges. Granted, much work has been done and continues to be done to nurture and recruit “culturally-connected” (being “Culturally Connected” entails one recognizing power dynamics in interaction, engaging proactively in learning, self-reflecting, and critiquing (cultural humility), promoting health literacy to develop a shared understanding of each other’s beliefs, values, priorities, and needs [6]) health care providers with (1) passion for work in rural communities; and (2) willingness to champion locally driven research and foster good decision making based on solid evidence [52]. The NBCFNB represents a model that ultimately aims to: generate evidence-based, local/rural healthcare research and solutions; produce effective and stable workforce capacity to operationalize solutions and fill the equity gap; and foster additional community benefit through accessing innovative research close to home. It also enables the inclusion of northern First Nations communities in a research domain that is typically southern and metropolitan-based and perhaps struggles in its own right for such diversity and generalizability. This is the call answered by the NBI, as it promotes partnerships that may improve healthcare and provides Indigenous students with mentorship opportunities to develop research skills.

Building research capacity within First Nations communities can yield long-term benefits related to both the depth and quality of the personnel in the research arena, including the capacity to: increase inclusion and promote culturally appropriate frameworks; select and shape research studies to address community priorities and health disparities; navigate the unique and complex issues of concern and relevant policy and ethical implications for research with First Nations communities; seek community input in shaping best practices for the dissemination of NBCFNB information in transparent, community-accessible formats; and lead responsive and relevant health research to influence future health promotion, policy development, and funding allocations in line with rural research-informed evidence. Having First Nations leaders and researchers in the NBCFNB building and implementation processes has the advantage of providing stronger voices in advocacy consultations for northern and rural health funding overall.

### 4.3. Developing Governance and Protocols That Support First Nations Peoples

While their position has been challenged by ongoing colonial policies, Indigenous Peoples are working to exercise governance over their data in accordance with traditional and cultural principles, perspectives, and practices [43,53]. The emphasis on NBCFNB consultations, the NFNBAC, and intergenerational transmission of Indigenous ways through Elders and Knowledge Keepers is not new.

Issues concerning the acquisition of biobanked tissue and data by researchers raise questions over the extent to which OCAP^®^ principles are applied. The real-world application of OCAP^®^ principles would allow First Nations Peoples to fully challenge dominant colonial and power-dynamic discourses, actively harness their ways of knowing and doing (via community-led governance mechanisms), and make appropriate decisions concerning the management of their NBCFNB. The First Nations governance structure that is planned in accordance with our consultations carries the role to ensure OCAP^®^ principles are respected.

### 4.4. Ensuring That Consent Is Appropriately Managed

In the First Nations context, it has been suggested that the collection of samples for a biobank is consented to both collectively and individually, with support from family and community leaders [54]. Since research on biobanked samples may impact family members, it is important that information concerning the nature of the research includes possible implications for this collective [55,56,57]. For full, ongoing informed consent to be culturally safe and promote research literacy, the use of preferred “culturally-connected” individuals is a vital consideration. With this in mind, despite the broad spectrum of First Nations’ languages and dialects in northern BC, there were limited requests for NBCFNB resources to be translated. Ensuring that terminology was used in explaining the NBCFNB in understandable terms was key. Communicating with pamphlets and other visual resources involving illustrations rather than significant text was another strong suggestion.

In line with this positive and communicative approach, participant concerns about being uninformed need continuous redress. At the time of consent, future research projects for which the de-identified samples or data in question may be used may not be known, meaning the magnitude of the risk or benefit may be vague. With the NBCFNB, the NBI team will continue to provide all necessary information pertaining to the NBCFNB while being cognizant of First Nations Peoples’ heterogeneity of culture, beliefs, and practices.

To address concerns about an NBCFNB implementation process, future misuse, and other aspects of the informed consent process is required. This is especially the case in this era of innovative technologies and advocacy for data sharing in order to enhance scientific advancement [58,59]. A First Nations-led discourse on expected ethical behavior in data management from program inception is mandatory. Reflecting on input, process to date, and literature addressing developments in the field of biobanking and Indigenous data sovereignty, such discourse may include respectful and responsive community engagement on details concerning potential benefits, practical aspects of maintaining privacy and confidentiality, appropriate pre-publication consultation with respect to research results by those granted access to an NBCFNB, and post-publication feedback commitments to the community [60].

Given the common research emphasis on maximizing data use through sharing—“open data”—ethical and legal issues are expected as the privacy and protection of Indigenous personal information in research are challenged [55]. Addressing these issues in small communities requires considering the potential benefits of participation in the NBCFNB against the possibility of identification, stigmatization, and emotional distress at the individual and community levels. As the governance protocols unfold, priority consideration must be given to building capacity for culturally appropriate communication in the consenting process that meets individual and community needs. To build a sense of collaboration and minimize emotional stress, informed and unbiased consent needs to be sought in a timely manner by trusted, trained individual(s) who are preferably First Nations or have experience working with First Nations and who establish trust with participating communities and their members.

A form of dynamic consent may be considered for implementation. If some prefer, the option of re-consenting may be explored and implemented in the consent process. The governance committee will assist in this next step of developing the consent process in detail from the perspectives of communities, leaders, and Elders/Knowledge Keepers. Elements to consider include the challenges of time to the individual participant, ongoing updates of contact information, implementation of resources, and protocols if the participant is deceased. Dynamic consent could be assisted through the internet or hand-held technologies, recognizing barriers to their use, particularly in participating remote northern regions. Garrison and colleagues contend that bi-directional engagement, increased communication, and transparency, with regards to potential proposals where existing samples may be used, could help maintain accountability and trust between individuals, their communities, and researchers and foster Indigenous control for individuals over potential uses of their samples and data [61]. Community members and leaders indicated the need for ongoing communication around the NBCFNB, including updates on research projects and publications at the very least through the standard process of FNHA communication: community newsletters, sub-regional and regional meetings. The role and impact of the emerging NBCFNB governance body will depend in part on the consent process selected through ongoing dialogue and the next steps.

The issue of sample repatriation remains complex and necessitates ongoing, open discussion. For some First Nations individuals, it may be unacceptable to not receive back a sample of sacred tissue from an ancestor once they have participated in a biobank. This belief is based on the existence of a close symbiotic relationship between Indigenous culture and the land [62]. The Traditional and ancestral land that First Nations Peoples have the right to protect from interference [63,64] is linked with their wellbeing, their waterways and wildlife, their dream stories, and their cultural knowledge. To respect these understandings and beliefs, some cultures advocate for the withdrawal or repatriation of samples for culturally based treatment or handling by Elders, communities, or families [50,65]. At a governance level, the overarching principle is that biobanked samples are “on loan”, to be protected, used responsibly, and returned upon request [66]. For the NBCFNB, the samples in question will be housed in the UHNBC under the stewardship of Northern Health. Input from First Nations communities will be critical in developing relevant governance guidelines on the vital issue of repatriation.

The discussion on the management of consent for the NBCFNB remains incomplete without consideration of what constitutes true reconciliation. With the recent passing of the BC Declaration on the Rights of Indigenous Peoples Act (*Declaration Act*) and Bill C-15 into law by both the Government of Canada and the Province of British Columbia [67,68], invalidation of Indigenous rights and processes and, within health research, usurpation and external ownership and control can be redressed.

### 4.5. Limitations

The NBI team acknowledges that self-rated knowledge in any qualitative study, including the NBCFNB consultations, can carry bias. Rating mechanisms are often driven subconsciously by researchers and thus difficult to interpret on the ground [69]. Participants may also feel a natural inclination toward increased scores with each rating or an expectation from researchers for their ratings to increase over time. One method for correcting this bias is to follow up with a measurable assessment, such as a quiz. For several reasons, we decided against this. First, our purpose for asking the question on biobank knowledge had more to do with understanding how participants felt their knowledge had changed than with a quantitative assessment of knowledge. Second, in an age where First Nations Peoples can feel over-researched, residential school experiences may have led to higher rates of anxiety, and relationship-building is important to increase trust and respect in the researcher–participant relationship, a quantitative assessment of knowledge was considered inappropriate. With guidance from the NFNBAC, our team and those supporting it felt that such an assessment could result in the accrual of undue stress on community members [70].

While not a limitation, the NBI team and FNHA Northern Team leadership recognize that this consultation process over the past nine years has provided only a foundation upon which to construct the recommended governance structure. From here, the proposed Governance Council will be created with direction from NFNBAC, the FNHA Northern Team leadership, and the appropriate protocols within the FNHA established by the Chiefs of the member First Nations. Once created, the subsequent members will draft Terms of Reference based on feedback received to date, and the resources to start addressing the outstanding questions and next steps will be provided.

## 5. Summary

While knowledge about biobanking increased following NBI information sessions, initially low levels of understanding around biobanking have important implications for future stages of NBCFNB consultation and development. The NBI will need to consider and address the contextual circumstances identified to increase community awareness and understanding of the nature, risks, and benefits of the NBCFNB and foster sustained engagement with the communities it is intended to serve. The NFNBAC and NBI team share the perspective that “dialogue-to-permit dialogue” and consultations with First Nations are important elements in the cycle of continuous conversation [44]. Importantly, the NBI information sessions and FGs represent the groundwork [34]. Now that a foundational dialogue has been established, opportunities to build upon the community engagement and health research literacy of First Nations in northern BC can follow.

During these consultations and the engagement of First Nations’ worldviews and community-specific values, the NBI team respectfully collaborated in the process, further understanding the historical and present contexts in which they continue to work. This partnership produced meaningful results that were accountable to and reflective of participants’ perspectives. This qualitative research will become a critically important basis upon which to inform culturally safe participation in genomic and genetic research, and its’ implications on healthcare, with the aim of improving patient outcomes for First Nations in northern BC and Canada. The ethical, legal, and social implications of genomic research must be understood within the local Indigenous context to foster the opportunities desired by communities and the health system at large. This study is the first known attempt to document the sociocultural values of genomic research in First Nations communities in BC. Understanding these values is important not only from a cultural safety perspective but also from a program sustainability point of view and avoiding any potential improper collection and use of biospecimens or data [71,72,73]. This is especially relevant in BC and Canada, with provincial and federal legislation implementing the United Nations Declaration on the Rights of Indigenous People.

This manuscript outlines a process by which First Nations communities throughout northern British Columbia were engaged regarding their opinions on a Northern BC First Nations Biobank. Individual interviews and focus groups were selected as mediums of engagement for several reasons. First, they are suitable for exploratory research. They are methods of choice when a topic has received limited attention in the literature and when it is necessary to explore firsthand knowledge from engaged individuals. Second, they are well suited to studies of human interaction. This is particularly the case with social research focusing on the dynamic interaction of individuals across different settings and situations, such as a health research environment or a biobank. Third, face-to-face engagement was considered appropriate and necessary to engender commitment among the communities and individuals that the NBCFNB is intended to serve. The interviews and focus groups yielded insights that will guide the establishment and operation of the NBCFNB, its advisory and governance, and the bilateral communication that must exist as an ethical space between researchers and primary stakeholders. Most importantly, the NBCFNB Advisory Committee recommended and facilitated these approaches.

The key principles and best practice approaches in this study are offered as considerations for those intending to begin their own processes of co-design with First Nations and other Indigenous communities. It is important to note, however, that each process requires a tailored approach that is grounded in the specific needs, characteristics, values, and cultures of the communities and individuals involved.

## Figures and Tables

**Figure 1 ijerph-20-05783-f001:**
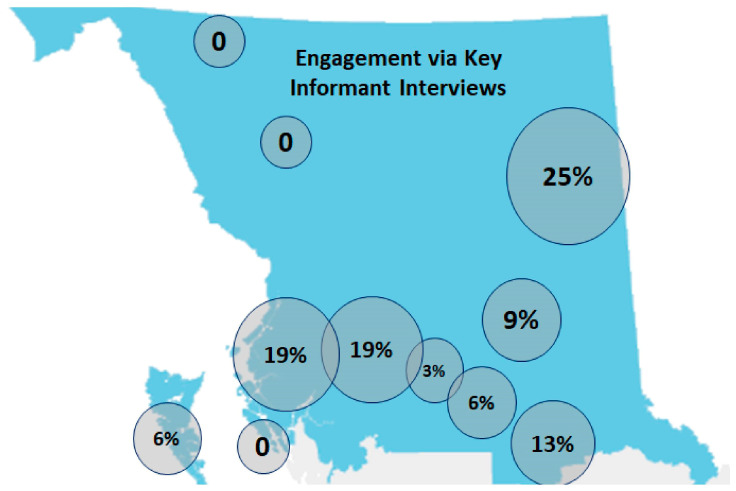
First Nations health authority engagement areas map showing Key Informant Interviews of First Nations in northern BC. Percentage of Key Informants [N = 32] from each of the 11 FNHA Engagement Areas: [Clockwise from top right.] Northeast: Treaty 8 Territory (25%); North Central: Finlay Hub (9%), Carrier South (13%), Carrier Lakes (6%), Lake Babine (3%); Northwest: Gitxsan & Wit’suwit’en (19%), Coast Mountain Alliance (19%), Coast Tsimshian (N = 0), Haida Gwaii (6%), Tahltan-Iskut, Dease Lake and Telegraph Creek (N = 0), True North (Kaska-Tlingit) (N = 0).

**Figure 2 ijerph-20-05783-f002:**
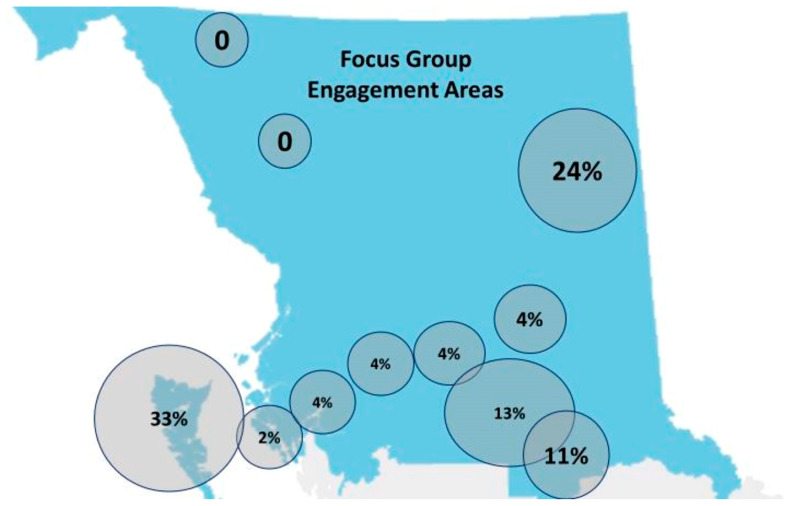
First Nations health authority engagement areas map showing focus groups of First Nations in northern BC. Percentage of focus group participants from each of 11 FNHA Engagement Areas [FG = 7; N = 46]: [Clockwise from top right.] Northeast: Treaty 8 Territory (24%); North Central: Finlay Hub (4%), Carrier South (11%), Carrier Lakes (13%), Lake Babine (4%); Northwest: Gitxsan & Wit’suwit’en (4%), Coast Mountain Alliance (4%), Coast Tsimshian (2%), Haida Gwaii (33%), Tahltan-Iskut, Dease Lake and Telegraph Creek (N = 0), True North (Kaska-Tlingit) (N = 0). Locations of the 7 NBI focus groups: Fort Nelson, Fort St. John, Prince George, Burns Lake, Terrace, Old Massett, Skidegate. The Dease Lake focus group was canceled due to forest fires. Community members participated in focus groups that were most accessible to them in terms of proximity and scheduling. Note: Percentages add up to 99% due to rounding decimals to the nearest 1.

**Figure 3 ijerph-20-05783-f003:**
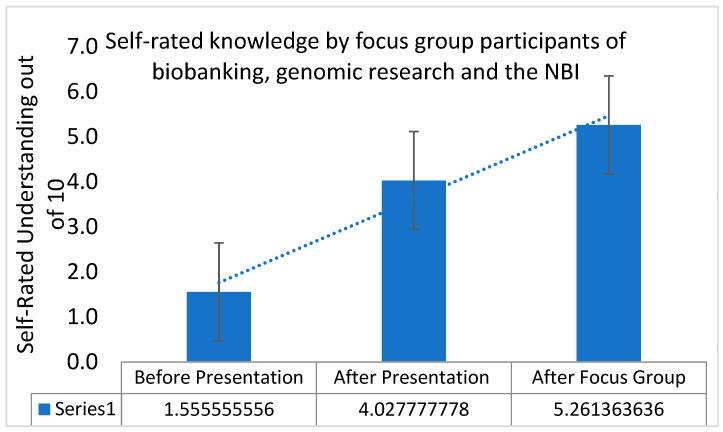
Focus group participants’ self-rated knowledge of biobanking, genomic research, and the Northern Biobank Initiative.

**Figure 4 ijerph-20-05783-f004:**
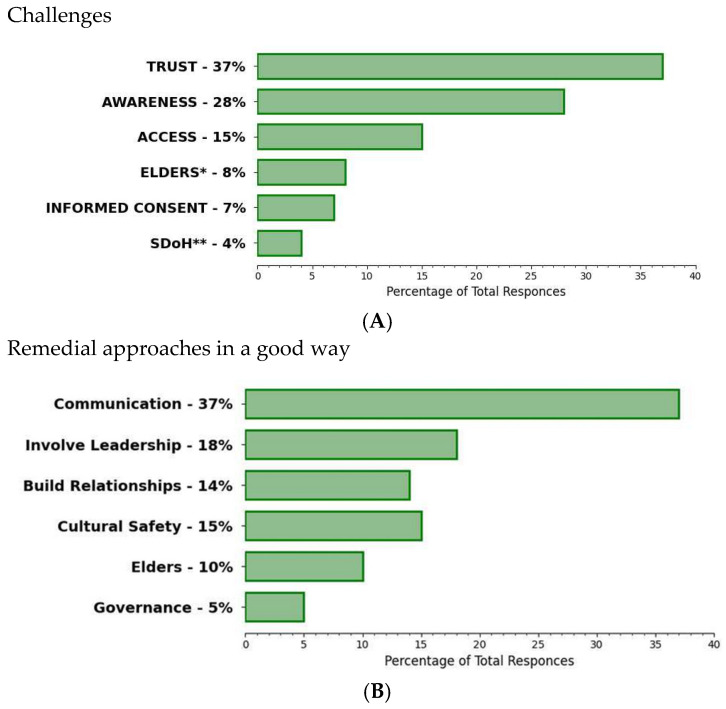
(**A**,**B**) Key informants: challenges of First Nations participation in biobanking and genomic research and remedial approaches. * Elders’ perspectives given history of Indian residential schools, Indian hospitals, unethical research, and their perspectives regarding associated research. ** SDoH = social determinants of health.

**Figure 5 ijerph-20-05783-f005:**
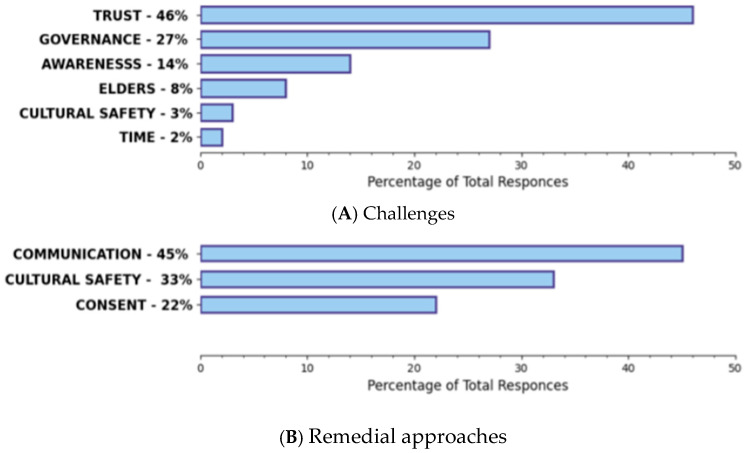
Community members: (**A**) challenges of First Nations participation in biobanking and genomic research and remedial approaches (**B**).

**Figure 6 ijerph-20-05783-f006:**
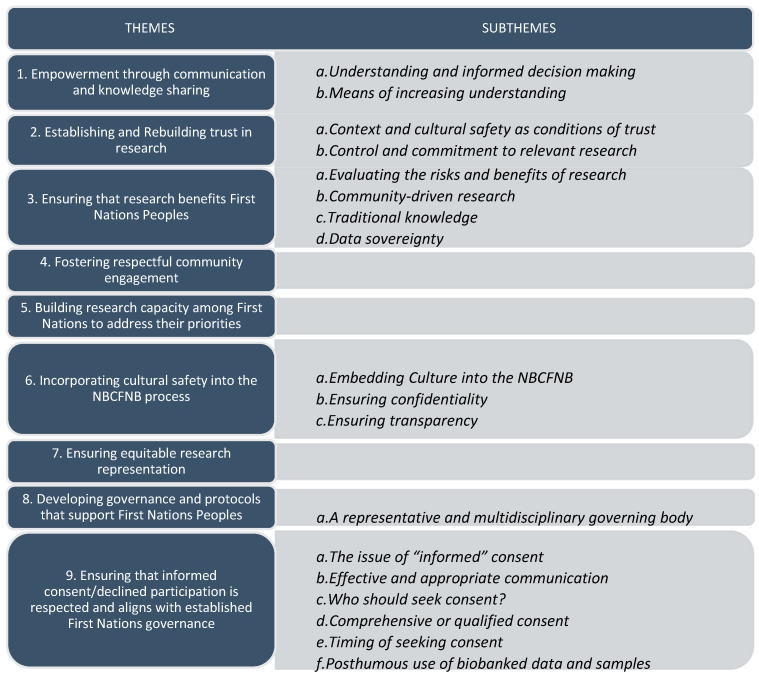
Themes and subthemes regarding the concept of a Northern BC First Nations biobank.

**Figure 7 ijerph-20-05783-f007:**
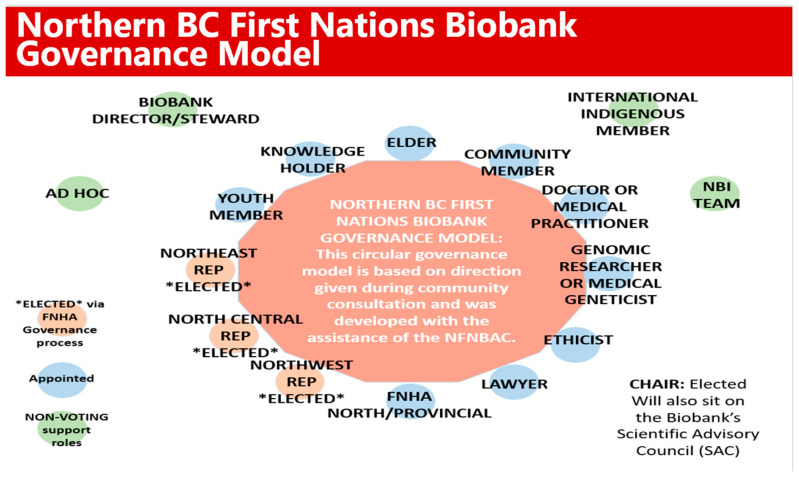
Proposed NBCFNB governance structure. *ELECTED*: representatives from each of the 3 sub-regions of northern BC, elected by First Nation leadership in their respective area and holding responsibilities that will be clarified and formalized during the establishment of the Governance Table and TOR.

**Table 1 ijerph-20-05783-t001:** The six phases of the Northern Biobank Initiative (NBI).

Stages	Initiation	Planning	Implementation and Execution			
Phases	Phase I	Phase II	Phase III	Phase IV	Phase V	Phase VI
Deliverable	- Business plan - Establish collaborations - Connect with the Canadian Tissue Repository Network (CTRNet) - Preliminary consultations	REB Approvals: ▪ Retrospective samples and database development ▪ Consultations and engagement Establish governance Sustainability plan	Prospective sample collection (breast, colon, thyroid, and melanoma cancers)	Fresh tissue sample collection	Integration of other tumour types to Northern Biobank Initiative (NBI)	Expansion of sample collection throughout the Northern Health Authority (NHA)

Note: Phases IV–VI may change order based on input from the established First Nations governance and Northern Biobank Initiative strategic development which is currently underway.

## Data Availability

The data presented in this study are available on request from the corresponding contact authors. The data are not publicly available due to confidentiality, privacy and ethical considerations.

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
