# Peer review of "Partnering with First Nations in Northern British Columbia Canada to Reduce Inequity in Access to Genomic Research"

_ijerph, 2023, doi:10.3390/ijerph20105783_

Round 1

Reviewer 1 Report

The manuscript entitled “Partnering with First Nations in Northern British Columbia Canada to Reduce Inequity in Access to Genomic Research” by Nadine Caron et al. presents the results from a phase of a very interesting, necessary and innovative initiative in the field of biobanking. It must be noted the high societal value of the project to minimize inequity of specific populations in access to research, and to show the point of view of First Nations and the solutions proposed based on a governance model.

Minor issues to be considered by the authors are:

-          Since the international scope of the article, abstract should be completed with a description of First Nations in order to understand the implications of the project and by who and how it is promoted.

-          Sentence “While the technologies…” in lines 77-80 could include the reasons explaining the differences in benefits of these two communities.

-          Last paragraph of Background should describe the project in a detailed way (table 1) and the context in which it is developed, to clearly understand the contributions of the results found.

-          Numerated sub-headings are necessary in sections 2, 3 and 4.

-          As a suggestion, quotes could be included as Supplementary results or annexes to improve the organization of the article. Consequently, structure of the article should be revised.

-          Discussion should be extended including results from similar initiatives and comparing the results found.

-          A section including abbreviations is missing.

Author Response

Dear esteemed reviewer,

We would like to express our gratitude for the suggestions and considerations put forth regarding our draft manuscript, Partnering with First Nations in Northern British Columbia Canada to Reduce Inequity in Access to Genomic Research.

We have reviewed each suggestion and addressed as indicated below.  In the few cases that the authors were not able to address issues, explanations are included. 

_ _ _

Since the international scope of the article, abstract should be completed with a description of First Nations in order to understand the implications of the project and by who and how it is promoted.

Done (limited), footnote.

Sentence “While the technologies…” in lines 77-80 could include the reasons explaining the differences in benefits of these two communities.

Done

Last paragraph of Background should describe the project in a detailed way (table 1) and the context in which it is developed, to clearly understand the contributions of the results found.

Done. 

Numerated sub-headings are necessary in sections 2, 3 and 4.

Done

As a suggestion, quotes could be included as Supplementary results or annexes to improve the organization of the article. Consequently, structure of the article should be revised.

Respectfully declined.  We strongly feel the importance of including the voices of the First Nations participants is paramount and must be included in the main body of the text, not as supplementary results or annexes.  This was an expectation of the Northern BC First Nations Biobank Advisory Committee with respect to honouring voices of community and leaders in this project.  It is critical that the quotes and voices be included in each section and read and considered, as they are, by the reader.

Discussion should be extended including results from similar initiatives and comparing the results found.

Done- see Discussion, relevant initiatives are considered and referenced. On March 3, 2023, a literature/research review was done and there was found to be no publications regarding Indigenous community consultation and the development of an Indigenous population-specific biobank.    

A section including abbreviations is missing.

Abbreviations are addressed as requested by the journal. 

_ _ _

Thank you again for your careful consideration and the improvements you have offered.  We look forward to the next steps.

In gratitude,

Shannon Hall

(on behalf of the submitting authors)

Reviewer 2 Report

The research tackles important topic of the importance of engagement of Indigenous Peoples in biobank in Northern British Columbia, Canada. I find the paper interesting, important and timely, especially that there is a scarcity of previous work on the topic. Another advantage of this study is the qualitative approach which enabled the Authors to describe and understand the meaning participants gave to their opinions on biobanking and genomic research. This in turn, provided new in-depth knowledge about the Indigenous Peoples experiences and views. However, while I believe that this research fills the gap in the literature and can be of interest to the readers of the Journal there are some issues that have to be revised before it could be published. Below I list the main points:

1. While the Journal has no word limit in my opinion the paper is a bit too long.

2. While according to the Journal’s requirements the Abstract should ‘be a total of about 200 words maximum’ it should highlight the purpose of the study, describe briefly the main methods applied, summarize the article's main findings, and indicate the main conclusions. In its present form the Abstract does not contain information on the methods used and should be more informative on the study results.

3. To contextualize the research in the Introduction section a more detailed information on a situation of Indigenous Peoples in Canada could be given, especially in relation to (genetic) research.

Moreover, for readers who are not familiar either with Canadian context or Indigenous Peoples some explanation on the concept of Elders/Knowledge Keepers is required (either in the text or as a footnote).

4. What, in my opinion, would be worth working on is the structure of the methodological part. For better reception, it is worth dividing the content of the methodological part according to the following scheme: study design, participants and setting, research tools, data collection, ethical issues, data analysis. In its present form, all these aspects are mixed, which reduces the methodological value of the work.

Additionally, the recruitment and data analysis are not totally clear.

5. A Table (Table 1. Study participants) with detailed information on study participants is required, following the scheme: code, gender, age, ethnicity, …

6. For better reception of the Results, apart from enumerating themes identified the Authors should prepare a thematic map showing main themes and subthemes as a separate figure.

7. lines 370-374: either delate (suggested) or move to the Methods section

8. No information on ELSI related to biobank research is given. See:

-- Bledsoe MJ. Ethical, legal and social issues of biobanking: past, present, and future. Biopreserv Biobanking, 2017;15(2):142–47.

--Domaradzki J. Geneticization and biobanking. Pol Sociol Rev. 2019;1(205):103–17.

--Caulfield, T. & Murdoch, B. Genes, cells, and biobanks: Yes, there’s still a consent problem. PLoS Biology. 2017;15(7):e2002654. https://doi.org/10.1371/journal.pbio.2002654.

9. Previous research demonstrated that biobanking and genetic research provoke some serious concerns among general population, and ethnic minorities. Thus, the Authors should describe that the barriers that hinder participation of Indigenous Peoples in biobanking and genomic research. For example, it should be noted that one of the reasons for lower level of trust toward science and state-sponsored research program among ethnic minorities, including African-Americans, Mexican-Americans, native Americans, Hawaii and Alaskan Natives, often results from their previous experiences with unethical healthcare research in ethnic populations (colonization, eugenics and medical experiments), their experiences with systemic racism and discrimination, under-representation of minorities in health research or negative experiences within a culturally insensitive healthcare system. For example, The Human Genome Diversity Project which intended to record the genetic profiles of indigenous populations was widely objected and accused of scientific racism, (bio)colonialism and biopiracy. See:

--Cavalli-Sforza LL. Opinion: the Human Genome Diversity Project: past, present and future. Nature Reviews Genetics. 2005;6(4): 333–340.

--Marks J. Human Genome Diversity Project (HGDP): impact on indigenous communities. In:  Clarke A, Ticehurst F.(Eds.), Living with the genome. Ethical and social aspects of human genetics, New York: Palgrave MacMillan, 2006, pp. 49–55.

--Aramoana J, Koea J, and on behalf of the CommNETS Collaboration. An Integrative Review of the Barriers to Indigenous Peoples Participation in Biobanking and Genomic Research . JCO Global Oncology 2020 :6, 83-91

-- Elsum I, McEwan C, Kowal EE, Cadet-James Y, Kelaher M, Woodward L. Inclusion of Indigenous Australians in biobanks: a step to reducing inequity in health care. Med J Aust. 2019;211(1):7-9.e1. doi: 10.5694/mja2.50219.

--Sinclair KA, Muller C, Noonan C, Booth-LaForce C, Buchwald DS. Increasing health equity through biospecimen research: Identification of factors that influence willingness of Native Americans to donate biospecimens. Prev Med Rep. 2021;21:101311. doi: 10.1016/j.pmedr.2021.101311.

--Mc Cartney AM, Anderson J, Liggins L, Hudson ML, Anderson MZ, TeAika B, Geary J, Cook-Deegan R, Patel HR, Phillippy AM. Balancing openness with Indigenous data sovereignty: An opportunity to leave no one behind in the journey to sequence all of life. Proc Natl Acad Sci U S A. 2022;119(4):e2115860119. doi: 10.1073/pnas.2115860119.

10. As for the Conclusions I suggest to discuss and give a more critical judgement on possible application of the results of the study. The Authors could also reflect more on the policy implications of their research: what solutions should be implemented in order to overcome the problem discussed in the manuscript. Thus, the paper would benefit from adding some recommendations suggesting possible guidelines for policymakers and/or researchers involved in biobank/genetic research.

Minor:

1. References enumeration needs to be revised: 8a, 11a, 27a-c, 30a-d?

2. “Indigenous Peoples” could be abbreviated through the text, for example IP.

3. Generally, shorter works quotes go in quotations marks within the text, and longer quotations are italicized. Thus, there is no need to highlight respondents quotations both using italics and quotations marks.

4. I am not sure whether all figures are necessary (i.e. 1B, 2B). Additionally, as figures 4a and 5a (but also 4b and 5b) address similar issues they could be put together so it would facilitate comparisons.

5. Minor spell check is required, i.e. ‘northern British Columbia’ (line 22) vs ‘Northern British Columbia’ (line 27)

Author Response

Dear esteemed reviewer,

We would like to express our gratitude for the suggestions and considerations put forth regarding our draft manuscript, Partnering with First Nations in Northern British Columbia Canada to Reduce Inequity in Access to Genomic Research.

We have reviewed each suggestion and addressed as indicated below.  In the few cases that the authors were not able to address issues, explanations are included. 

 _ _ _

While the Journal has no word limit in my opinion the paper is a bit too long.

The authors understand this comment, but in this instance, have declined to change due to scope of paper, importance of topic, and the necessity of giving full consideration and voice to the partners and people involved in the study and the work. While efforts were made to shorten the paper, addressing reviewer’s requests balanced any success of shortening the paper without losing the depth of Indigenous community voice.

While according to the Journal’s requirements the Abstract should ‘be a total of about 200 words maximum’ it should highlight the purpose of the study, describe briefly the main methods applied, summarize the article's main findings, and indicate the main conclusions. In its present form the Abstract does not contain information on the methods used and should be more informative on the study results.

Done

To contextualize the research in the Introduction section a more detailed information on a situation of Indigenous Peoples in Canada could be given, especially in relation to (genetic) research.

Authors felt that sufficient information was provided, particularly since the focus of this paper is solely on the First Nations peoples in northern British Columbia.  This is a qualitative study to learn about the perspectives and understanding about biobanks with these communities and their leadership.  The request of the reviewer is a great idea for a follow up manuscript.  

Moreover, for readers who are not familiar either with Canadian context or Indigenous Peoples some explanation on the concept of Elders/Knowledge Keepers is required (either in the text or as a footnote).

Done

What, in my opinion, would be worth working on is the structure of the methodological part. For better reception, it is worth dividing the content of the methodological part according to the following scheme: study design, participants and setting, research tools, data collection, ethical issues, data analysis. In its present form, all these aspects are mixed, which reduces the methodological value of the work.

Done

Additionally, the recruitment and data analysis are not totally clear.

Done

A Table (Table 1. Study participants) with detailed information on study participants is required, following the scheme: code, gender, age, ethnicity, …

Additional information on recruitment and general information on study participants provided in text; a table is not provided due to the nature of recruitment, confidentiality,  and engagement principles.

For better reception of the Results, apart from enumerating themes identified the Authors should prepare a thematic map showing main themes and subthemes as a separate figure.

Done

Lines 370-374: either delate (suggested) or move to the Methods section

Declined.  Authors feel these are results rather than methods, and are important learnings.

No information on ELSI related to biobank research is given. See:

-- Bledsoe MJ. Ethical, legal and social issues of biobanking: past, present, and future. Biopreserv Biobanking, 2017;15(2):142–47.

--Domaradzki J. Geneticization and biobanking. Pol Sociol Rev. 2019;1(205):103–17.

--Caulfield, T. & Murdoch, B. Genes, cells, and biobanks: Yes, there’s still a consent problem. PLoS Biology. 2017;15(7):e2002654. https://doi.org/10.1371/journal.pbio.2002654.

Done

Previous research demonstrated that biobanking and genetic research provoke some serious concerns among general population, and ethnic minorities. Thus, the Authors should describe that the barriers that hinder participation of Indigenous Peoples in biobanking and genomic research. For example, it should be noted that one of the reasons for lower level of trust toward science and state-sponsored research program among ethnic minorities, including African-Americans, Mexican-Americans, native Americans, Hawaii and Alaskan Natives, often results from their previous experiences with unethical healthcare research in ethnic populations (colonization, eugenics and medical experiments), their experiences with systemic racism and discrimination, under-representation of minorities in health research or negative experiences within a culturally insensitive healthcare system. For example, The Human Genome Diversity Project which intended to record the genetic profiles of indigenous populations was widely objected and accused of scientific racism, (bio)colonialism and biopiracy. See:

--Cavalli-Sforza LL. Opinion: the Human Genome Diversity Project: past, present and future. Nature Reviews Genetics. 2005;6(4): 333–340.

--Marks J. Human Genome Diversity Project (HGDP): impact on indigenous communities. In:  Clarke A, Ticehurst F.(Eds.), Living with the genome. Ethical and social aspects of human genetics, New York: Palgrave MacMillan, 2006, pp. 49–55.

--Aramoana J, Koea J, and on behalf of the CommNETS Collaboration. An Integrative Review of the Barriers to Indigenous Peoples Participation in Biobanking and Genomic Research . JCO Global Oncology 2020 :6, 83-91

-- Elsum I, McEwan C, Kowal EE, Cadet-James Y, Kelaher M, Woodward L. Inclusion of Indigenous Australians in biobanks: a step to reducing inequity in health care. Med J Aust. 2019;211(1):7-9.e1. doi: 10.5694/mja2.50219.

--Sinclair KA, Muller C, Noonan C, Booth-LaForce C, Buchwald DS. Increasing health equity through biospecimen research: Identification of factors that influence willingness of Native Americans to donate biospecimens. Prev Med Rep. 2021;21:101311. doi: 10.1016/j.pmedr.2021.101311.

--Mc Cartney AM, Anderson J, Liggins L, Hudson ML, Anderson MZ, TeAika B, Geary J, Cook-Deegan R, Patel HR, Phillippy AM. Balancing openness with Indigenous data sovereignty: An opportunity to leave no one behind in the journey to sequence all of life. Proc Natl Acad Sci U S A. 2022;119(4):e2115860119. doi: 10.1073/pnas.2115860119.

Done to a degree amenable to the authors without dramatically increasing (or diverting) the focus or scope of the paper.

As for the Conclusions I suggest to discuss and give a more critical judgement on possible application of the results of the study. The Authors could also reflect more on the policy implications of their research: what solutions should be implemented in order to overcome the problem discussed in the manuscript. Thus, the paper would benefit from adding some recommendations suggesting possible guidelines for policymakers and/or researchers involved in biobank/genetic research.

Done (summary)

Minor:

References enumeration needs to be revised: 8a, 11a, 27a-c, 30a-d?

Done

“Indigenous Peoples” could be abbreviated through the text, for example IP.

Authors appreciate the suggestion, and after discussion with Indigenous colleagues, it was decided to keep the full phrase as it is more respectful.

Generally, shorter works quotes go in quotations marks within the text, and longer quotations are italicized. Thus, there is no need to highlight respondents quotations both using italics and quotations marks.

Corrected, quotation marks removed for italicized, indented longer quotations.

I am not sure whether all figures are necessary (i.e. 1B, 2B). Additionally, as figures 4a and 5a (but also 4b and 5b) address similar issues they could be put together so it would facilitate comparisons.

Done.  1B, 2B removed with sub-regional information added to 1A and 2A descriptions.  Authors retained remaining figures.

Minor spell check is required, i.e. ‘northern British Columbia’ (line 22) vs ‘Northern British Columbia’ (line 27)

One instance was corrected.  Remaining capitalized ‘Northern’s’ are capitalized as they are names. 

_ _ _

Thank you again for your careful consideration and the improvements you have offered.  We look forward to the next steps.

In gratitude,

Shannon Hall

(on behalf of the submitting authors)

Round 2

Reviewer 2 Report

I have read both all the reviews and Authors’ response and the revised manuscript itself with interest. The Authors have clarified all issues raised in the review and I believe that this revised manuscript is now more consistent owing to their corrections and additional arguments. On the whole, I appreciate this effort and have no further concern regarding the manuscript. All in all, I believe that the article is important and interesting and fits well with the aims of Journal. For that reason  recommend its publication.